# Metal–Organic Frameworks as Fillers in Porous Organic Polymer-Based Hybrid Materials: Innovations in Composition, Processing, and Applications

**DOI:** 10.3390/polym17141941

**Published:** 2025-07-15

**Authors:** Victor Durán-Egido, Daniel García-Giménez, Juan Carlos Martínez-López, Laura Pérez-Vidal, Javier Carretero-González

**Affiliations:** 1Faculty of Chemistry, Universidad Complutense de Madrid, 28040 Madrid, Spain; vduran01@ucm.es; 2Institute of Polymer Science and Technology, ICTP, CSIC, 28006 Madrid, Spain; danielgargi@ictp.csic.es (D.G.-G.); juancarml@ictp.csic.es (J.C.M.-L.); laurapervid@ictp.csic.es (L.P.-V.)

**Keywords:** metal-organic framework, porous organic polymer, covalent organic framework

## Abstract

Hybrid materials based on porous organic polymers (POPs) and metal–organic frameworks (MOFs) are increasing attention for advanced separation processes due to the possibility to combine their properties. POPs provide high surface areas, chemical stability, and tunable porosity, while MOFs contribute a high variety of defined crystalline structures and enhanced separation characteristics. The combination (or hybridization) with PIMs gives rise to mixed-matrix membranes (MMMs) with improved permeability, selectivity, and long-term stability. However, interfacial compatibility remains a key limitation, often addressed through polymer functionalization or controlled dispersion of the MOF phase. MOF/COF hybrids are more used as biochemical sensors with elevated sensitivity, catalytic applications, and wastewater remediation. They are also very well known in the gas sorption and separation field, due to their tunable porosity and high electrical conductivity, which also makes them feasible for energy storage applications. Last but not less important, hybrids with other POPs, such as hyper-crosslinked polymers (HCPs), covalent triazine frameworks (CTFs), or conjugated microporous polymers (CMPs), offer enhanced functionality. MOF/HCP hybrids combine ease of synthesis and chemical robustness with tunable porosity. MOF/CTF hybrids provide superior thermal and chemical stability under harsh conditions, while MOF/CMP hybrids introduce π-conjugation for enhanced conductivity and photocatalytic activity. These and other findings confirm the potential of MOF-POP hybrids as next-generation materials for gas separation and carbon capture applications.

## 1. Introduction

Metal–organic frameworks (MOFs) have emerged as one of the most versatile classes of porous materials, characterized by their tunable crystalline structures and high surface areas (500–7000 m^2^/g) [1,2,3]. Constructed through the coordination of metal nodes or metal–oxo clusters with organic linkers, MOFs exhibit remarkable potential in areas such as gas storage [4], separation [5], catalysis [6], energy [7], and sensing applications [8,9]. However, despite their structural and functional diversity, MOFs often face limitations such as moderate stability under harsh chemical conditions, poor processability, and challenges in integrating into practical devices [10]. These shortcomings have motivated researchers to combine them with other materials to form hybrids such as different polymer matrixes. For instance, the creation of hybrids made of MOFs and polymers leads to a broader spectrum of favorable properties than do the individual MOF or polymer components separately [11]. Combining polymers and MOFs with well-established functional properties could yield easily processable and ready-to-use formulations [12]. Diverse approaches have been explored by research groups where MOF−polymer composites comprise the leading MOF-based mixed-matrix membranes (MMMs) [13,14], matrix-free hybrids with polymers grafted from the surface of MOFs or through MOF particles [15], MOF-templated polymers where polymerization takes place in the free space of MOF channels [16], polymers templating MOF growth [17,18], and the synthesis of MOFs using polymer ligands (polyMOFs) [17,19].

Porous organic polymers (POPs) represent an ideal complementary material class for hybridization with MOFs. The combination of MOFs with POPs enhances the performance of the hybrid system compared to the individual counterparts and extends their applicability. POPs offer outstanding chemical and thermal stability due to their entirely covalent framework [20]. Their synthesis involves robust polymerization strategies such as condensation reactions, Suzuki coupling, Sonogashira coupling, and Friedel–Crafts alkylation, allowing for precise control over pore structure and functionality [21]. Porous organic polymers encompass a diverse range of materials, each with distinct structural and functional characteristics that make them suitable for hybridization with MOFs.

**Covalent organic frameworks (COFs)** stand out among POPs due to their crystalline structure, which provides well-defined, ordered porosity [22]. Their tunable pore sizes, high surface areas, and chemical versatility make them particularly suited for integration with MOFs in applications such as catalysis, gas separation, energy storage, and biosensors [23,24]. Another important group is **conjugated microporous polymers (CMPs)**, which combine permanent microporosity with extended π-conjugation [25]. This structure imparts electronic conductivity, making CMPs highly valuable in energy storage [26], photocatalysis [27], and optoelectronics [28]. Their conjugated nature also promotes strong π–π interactions with MOFs, enhancing interfacial adhesion in hybrid materials [29]. In contrast to the ordered networks of COFs and CMPs, **hyper-crosslinked polymers (HCPs)** are amorphous materials characterized by ultra-microporosity [30]. Typically synthesized via Friedel–Crafts reactions, they offer exceptional chemical stability and hydrophobicity, making them ideal for adsorption-based applications [31]. Their rigid frameworks also reinforce MOFs structurally, enhancing mechanical stability in hybrid systems [32]. **Covalent triazine frameworks (CTFs)** are another class of POPs, notable for their nitrogen-rich composition and robust structures [33,34]. Synthesized via the trimerization of aromatic nitriles, they exhibit strong interactions with metal sites, facilitating their integration with MOFs. The nitrogen functionalities of CTFs improve interfacial adhesion by coordinating with MOF metal nodes, enhancing hybrid stability and performance [35]. MOF–CTF hybrids are commonly synthesized by growing MOFs on CTF surfaces or embedding MOFs within CTF networks, resulting in hierarchical porosity that benefits applications in catalysis [36], separation [37,38], and energy storage [39]. **Polymers of intrinsic microporosity (PIMs)** differ from other POPs due to their rigid and contorted backbones, which prevent efficient packing and create permanent free volume [21]. This structure grants them high permeability and excellent gas transport properties. Their combination of porosity and flexibility makes PIMs valuable in MOF-based membranes for separation technologies, where they enhance processability and introduce complementary pore structures [40,41,42].

Therefore, the combination of MOFs and POPs has given rise to a rapidly expanding field of materials research that leverages the strengths of both classes. By integrating these two porous systems, researchers can achieve advanced materials with improved stability, enhanced processability, and synergistic functionality [43].

Beyond their advantages, however, MOF–POP hybrids also present challenges in terms of compromising MOF structure or accessible porosity [15]. One of the primary difficulties is achieving optimum interfacial compatibility between the two components [44]. Because MOFs are constructed by coordination bonds and POPs are covalent networks, ensuring sufficient adhesion and interaction at their interface is crucial [45,46]. Differences in chemical functionality, hydrophobicity, and thermal expansion behavior can lead to phase separation or weak interfacial contact, which may limit hybrid performance [47]. Additionally, while the structural diversity of MOFs allows for significant functionalization, maintaining their crystallinity during polymer integration can be challenging [48]. The synthesis of MOF–POP hybrids often requires precise control over reaction conditions to ensure homogeneous mixing, avoid framework collapse, and optimize interactions between the materials. Furthermore, while hybridization opens new avenues for enhanced performance, it may also increase synthetic complexity [12].

A crucial step in addressing these challenges is the strategic functionalization of MOFs, which enhances their compatibility with polymeric frameworks [49]. Modifications at the metal clusters or organic linkers can introduce functional groups that improve adhesion, promote covalent or non-covalent interactions, and optimize interfacial contact between MOFs and POPs. Likewise, tailoring the chemistry of POPs (e.g., incorporating polar groups or π-conjugated systems) further enhances hybrid formation [50], ensuring efficient charge or mass transfer across the interface. In the following sections, we will delve into the structural diversity of MOFs, highlighting their functionalization strategies and their role in facilitating MOF–POP hybridization. We will then discuss the synthetic methodologies employed in fabricating these hybrids, examine their physicochemical properties, and assess their potential applications in gas separation, catalysis, and energy storage and conversion.

## 2. Diversity of MOF Structures and Their Functionalization Potential

Metal–organic frameworks (MOFs) exhibit an extraordinary diversity of structures due to the nearly limitless combinations of metal nodes and organic linkers [51]. This tunability enables precise control over pore size, shape, and chemical functionality, making MOFs highly versatile for various applications [52]. The fundamental structural diversity of MOFs stems from the coordination geometry of metal centers and the connectivity of organic ligands, leading to distinct topologies. Among the vast array of MOF architectures, several common structural types have been extensively studied and employed for hybridization with porous organic polymers (POPs), some of which are summarized in Table 1 and further described below.

### 2.1. Zeolitic Imidazolate Frameworks (ZIFs)

Zeolitic imidazolate frameworks (ZIFs) represent a remarkable subclass of MOFs, blending the structural features of zeolites with the tunability of metal–organic frameworks. Their defining characteristic lies in the tetrahedral coordination of metal centers, usually Zn^2+^ or Co^2+^, with imidazolate linkers, resulting in a robust structure that withstands high temperatures and extreme pH conditions (pH 2–12). This extraordinary stability [53] makes ZIFs invaluable for gas separation, catalysis, and chemical sensing, particularly in environments where conventional MOFs may degrade [75]. Among the numerous ZIFs, ZIF-8 and ZIF-67 stand out as the most studied. ZIF-8, featuring zinc centers coordinated with 2-methylimidazolate, is highly hydrophobic and chemically resilient, making it ideal for selective gas adsorption and membrane technologies [53,76]. In contrast, ZIF-67, with cobalt in place of zinc, finds widespread application in electrocatalysis and energy storage [77,78].

One of the most compelling aspects of ZIFs is their ability to be functionalized, allowing fine-tuned properties for specific applications. The imidazolate linkers can be modified with functional groups like methyl (-CH_3_), nitro (-NO_2_), or amine (-NH_2_), which particularly enhances the pervaporation performance of ethanol/water separation processes [79]. This functionalization has a great influence on hydrophilicity, electronic properties, and interactions with other materials [80,81]. Such modifications are particularly beneficial when hybridizing ZIFs with porous organic polymers (POPs) [82]. Functionalized linkers can enhance adhesion through hydrogen bonding, π–π stacking, or acid–base interactions, ensuring a stable and uniform hybrid structure. These composites, which combine the chemical resilience of ZIFs with the processability of polymers, are gaining traction in membrane-based gas separation, hybrid catalysts, and next-generation sensors [83].

### 2.2. UiO-Series MOFs (UiO-66, UiO-67, UiO-68)

The UiO family of MOFs, named after the University of Oslo, is among the most chemically robust and thermally stable metal–organic frameworks [59]. Their exceptional durability arises from the strong coordination between zirconium clusters and carboxylate linkers, making them highly resistant to water, heat, and harsh chemical conditions. Unlike many MOFs, which suffer from structural collapse in humid environments [84], UiO materials maintain their integrity, making them ideal for water-stable catalysis [85], adsorption [86], and energy storage [87]. Among the widely studied UiO MOFs, UiO-66, UiO-67, and UiO-68 exhibit subtle yet important differences. UiO-66, constructed with terephthalic acid (BDC) linkers, is the most extensively investigated, owing to its high surface area and tunable porosity [88]. Expanding on this design, UiO-67 and UiO-68 feature longer linkers—biphenyl dicarboxylate (BPDC) and terphenyl dicarboxylate (TPDC), respectively—resulting in increased pore sizes and enhanced molecular diffusion [60].

A major advantage of UiO MOFs lies in their versatility in functionalization. The carboxylate linkers can be modified with amines (-NH_2_) whereupon an increase in ionic conductivity is observed, benefiting from the new positively charged sites [89], as well as an increase in basicity of the Zr_6_O_4_(OH)_4_ clusters, which allows for higher acid adsorption properties [90]. Other modifications, such as functionalizing with hydroxyls (-OH) or sulfonic acids (-SO_3_H), lead to significant alterations in their surface properties and compatibility with other materials [35]. These functional groups enhance hydrogen bonding, acid–base interactions, and even covalent bonding with polymers, making UiO MOFs excellent candidates for hybridization with POPs. Strategies such as post-synthetic modification (PSM) and linker exchange further refine their properties, enabling the formation of composite materials that combine high porosity with enhanced mechanical flexibility [14,41].

### 2.3. MIL-Series MOFs (MIL-101, MIL-53, MIL-100)

The MIL (Matériaux de l’Institut Lavoisier) family of MOFs, predominantly constructed from iron (Fe), chromium (Cr), aluminum (Al), or vanadium (V) clusters and polycarboxylates [91,92,93], is distinguished by its large pore volume, high surface area, and remarkable chemical adaptability. These properties make MIL MOFs exceptionally versatile for catalysis, gas adsorption, drug delivery [94], and energy applications [95]. Among the most renowned MIL MOFs, MIL-101 stands out due to its massive mesoporous cages (2.9 nm and 3.4 nm in diameter), which enable exceptional molecular diffusion and guest encapsulation [96]. Another important member, MIL-53, is known for its “breathing effect,” where its pores dynamically expand and contract in response to external stimuli, enhancing its performance in CO_2_ capture and adsorption-based applications [67].

To enhance their interaction with polymers, MIL MOFs can be post-synthetically modified with sulfonic acid (-SO_3_H), amine (-NH_2_), or hydroxyl (-OH) groups. These modifications improve hydrophilicity, chemical reactivity, and interfacial adhesion with POPs, ensuring better mechanical stability in composite materials. Hybridization strategies often exploit MIL-101’s large pores, which allow polymer chains to infiltrate the framework, resulting in highly stable interpenetrated hybrid structures with enhanced gas storage and catalytic properties [97].

### 2.4. HKUST-1 (Cu_3_(BTC)_2_, Copper–Benzene Tricarboxylate MOF)

HKUST-1 (where HKUST stands for Hong Kong University of Science and Technology) is one of the most recognizable MOFs, featuring copper-based Cu_2_ paddlewheel units bridged by benzene-1,3,5-tricarboxylate (BTC) linkers [98]. With its high porosity (~1800 m^2^/g) and excellent gas adsorption properties, it has been extensively utilized in gas storage [99,100], catalysis [101], and separation technologies [99]. However, a major limitation of HKUST-1 is its moderate stability in humid environments [84], as water molecules can coordinate with Cu^2+^ sites, leading to gradual framework degradation. To mitigate this, functionalization strategies involving hydrophilic or hydrophobic modifications are employed [43]. By modifying BTC linkers with hydroxyl (-OH) or carboxyl (-COOH) groups, HKUST-1 can also achieve better compatibility with polymers, improving interfacial adhesion and long-term stability.

Hybridization strategies often leverage ligand modification and direct coordination with polymer chains, ensuring that HKUST-1 retains its high adsorption capacity while improving mechanical strength [102].

### 2.5. MOFs with Pillared-Layer Structures (MOF-5, DUT-Series)

MOF-5, one of the earliest and most studied MOFs, consists of zinc oxide clusters coordinated with benzene dicarboxylate (BDC) linkers, forming a highly porous cubic framework [70]. This structure has inspired a range of derivatives with enhanced chemical stability and tunable porosity, including the DUT (Dresden University of Technology) series, which incorporates longer and more flexible linkers to optimize gas storage and separation properties [103].

Functionalization strategies (Figure 1), as with all other carboxylate-based MOFs, focus on modifying the BDC linkers with amine (-NH_2_), hydroxyl (-OH), or sulfonic acid (-SO_3_H) groups, so that compatibility with polymeric matrices can be improved [104], or combining different ligands (azolates and carboxylates) in the same structure [105], yielding high CO_2_ selectivity.

Therefore, functionalization plays a pivotal role in improving the compatibility between MOFs and porous organic polymers. The abovementioned covalent functionalization, where reactive groups such as amines, thiols, or sulfonic acids are introduced onto the MOF linkers, provides functional groups that can form strong covalent bonds with polymer chains, creating a more integrated and durable hybrid structure [106]. In addition to covalent bonding, non-covalent interactions can also play a crucial role in enhancing interfacial adhesion between MOFs and POPs. For instance, hydrogen bonding, π–π stacking, and electrostatic interactions help reinforce the hybrid interface, improving the overall mechanical integrity of the material while also maintaining high porosity [48].

Some widely used techniques for this functionalization are post-synthetic modification (PSM) and ligand exchange. Whereas the first allows for the introduction of functional groups onto the MOF framework after its initial synthesis and provides greater flexibility in tailoring the surface chemistry of MOFs to make them more compatible with polymer networks (thereby facilitating stronger interactions and better integration), in the latter some of the original MOF linkers are replaced with new, polymer-compatible moieties [107]. This modification enhances the hybridization potential by introducing linkers that are more chemically suited to interacting with POPs, all while maintaining the structural integrity and porosity of the MOF. In the following sections, we will focus on the latest advances in synthetic approaches and applications in MOF-POP hybrids based on the principal families of these porous organic polymers.

## 3. MOF-PIM Hybrids (MMMs)

PIMs are a specific type of POP with some specific properties that make them very suitable for making hybrid materials [42,108]. These polymers comprise fused rings and a contorted structure that leads to an inefficient packing of the chain, giving rise to sub-nanometer micropores and high fractional free volume, leading to Brunauer–Emmett–Teller (BET) surface areas of 500–800 m^2^·g^−1^. Moreover, thanks to the not-crosslinked structure of these polymers, they are able to generate microporous membranes without any activation or modification and using simple techniques [109], which makes them very attractive for gas separations or solvent purification purposes and for industrial application, since they can be processed into films by simple techniques like film casting [110,111,112].

While MOFs and PIMs have some disadvantages when are used separately, like aging and the low gas selectivity of most PIMs, and the non-processability of MOFs, their combination can lead to enhanced properties, increasing the lower separation efficiency of the polymers and allowing inorganic fillers to be processed into scaled-up materials, leading to advances in fields like gas separation, CO_2_ adsorption, catalysis, or energy applications (Figure 2) [113,114].

Hybrid materials combining MOFs and PIMs are usually used as membranes, called mixed-matrix membranes (MMMs). These membranes were first developed by Paul and Kemp in 1973 using zeolite and silicon as components [115]. They are usually formed by a filler and a polymer matrix. Due to their properties, MOFs are very versatile fillers for MMMs, since their high porosity and gas permeability allow them to be used in separation processes for industry application [115]. Moreover, these fillers need a polymer matrix with good performance in gas separation and easy processability. PIMs can be processed in solution by using a wide range of solvents and have a high gas permeability, which makes them a good candidate to combine with MOFs.

However, there are some challenges in the fabrication of MMMs. For instance, PIMs are chemically and mechanically stable, but they are prone to aging, which can reduce permeability efficiency and increase selectivity over time due to chain relaxation over time [113,116]. Hybridizing PIMs with MOFs mitigates these drawbacks while enhancing performance [117,118]. Moreover, MOFs restrict polymer chain mobility, reducing swelling and plasticization while preventing long-term aging [119,120].

In gas separation application, the incorporation of highly selective fillers can enhance permeability by increasing the free volume in the membrane and improving the selectivity, breaking with the upper bound trend [121], where normally one of them increases and the other one decreases [122]. However, two of the main problems of MMMs using PIMs are the interface compatibility with the inorganic filler, leading to gaps, poor dispersion, and compromised selectivity and permeability, which can limit the performance of the membrane [123,124], and the particle aggregation, which can lead to inhomogeneity in the hybrid material due to the MOF particle agglomeration in the polymer matrix and poor mechanical properties.

There are some strategies to follow to solve these problems. On the one hand, introducing functional groups like a hydroxyl group (OH) between filler and polymer can improve adhesion, but does not always work as desired [50]; MOFs can be incorporated into polymer matrices using noncovalent interactions such as hydrogen bonds or π–π stacking, which is achieved by dispersing MOF ink into a polymer solution and casting films; or more flexible polymers or even a combination can be chosen, like some PIM–polyimides, which are more flexible but can also create hydrogen bonds with MOF fillers like ZIF-8 [125,126]. On the other hand, to mitigate particle aggregation, optimized dispersion techniques and covalent bonding can improve uniformity and homogeneity.

Taking into account the strategies to improve the interaction between filler and polymer, different fabrication methods have been developed, but two of them are the most used ones:**Solution mixing:** The polymer is first dissolved in a suitable solvent, and the MOF filler is dispersed in the same solvent, often using ultrasonication to achieve a homogeneous mixture. This approach is simple and widely applied; however, it may involve risks such as partial loss of MOF crystallinity, especially under prolonged ultrasonication or in the presence of aggressive solvents, which can compromise the structural integrity of the framework and reduce its adsorption performance. Similarly, depending on the polymer chemistry, some POPs may undergo chemical degradation or swelling during extended exposure to polar solvents or elevated temperatures used in casting [127,128].**In situ method:** This involves mixing the filler (typically pre-formed MOFs) with monomers prior to polymerization, allowing the filler to become embedded within the growing polymer network. A variation of this approach is in situ MOF growth, where MOF precursors are added to a pre-formed polymer solution, and crystallization is induced within the polymer matrix. This technique often enhances MOF dispersion and interfacial compatibility but requires precise control of synthesis conditions to avoid defective MOF growth or unintended interactions with functional groups in POPs, which might limit polymer performance or porosity development. In some cases, acidic or basic by-products generated during MOF crystallization can disrupt the polymer structure, especially in POPs containing hydrolytically sensitive linkages such as imines or boronate esters. Moreover, the thermal and chemical conditions required for successful MOF nucleation (such as solvothermal treatment or the presence of modulators) may exceed the stability window of certain polymers, resulting in chain scission, crosslinking, or partial collapse of the porous architecture. Therefore, careful optimization of the precursor ratios, reaction time, and solvent environment is essential to preserve both the structural integrity of the MOF and the intrinsic microporosity of the polymer phase [124,127,128].

### Applications of MOF-PIM MMMs

MMMs are versatile composite materials that can be used in many fields, due to the advanced properties obtained from the combination of the filler and the polymer matrix. In the case of MMMs made from MOF–filler and PIM polymers, the main applications are in the gas separation and CO_2_ capture fields, but others, like hydrogen separation, natural gas purification, organic solvent filtration, and propylene separation, are also described, as can be seen in Figure 3 [105,126,129,130,131,132,133,134,135,136,137,138].

The MMM family formed from CAU-12-ODB and PIM-1 is an example of a hybrid material used in **hydrogen permeability applications**. Chi Zhang et al. reported a very selective MMM with an H_2_/N_2_ selectivity of 127 and a P_H2_ of 7199 Barrer (Barrer is the typical unit used to quantify gas permeability in polymer membranes. It is defined as the amount of gas (in cm^3^, at standard temperature and pressure) that passes through a membrane of 1 cm thickness and 1 cm^2^ area per second, under a pressure gradient of 1 cmHg. Mathematically, 1 Barrer = 10^−10^ cm^3^ (STP)·cm/(cm^2^·s·cmHg). This unit integrates both the diffusivity and solubility of the gas in the membrane, making it a fundamental parameter in characterizing gas transport properties). CAU-12 filler has a pore diameter of 3.3 Å, just between the kinetic diameters of H_2_ and N_2_ (2.9 and 3.64, respectively), which, together with the high permeability of the PIM-1 and the good interfacial adhesion between them, makes them very suitable for hydrogen separation [129].

Shaohui Xiong et al.’s research is another example of using MMM hybrid materials for a hydrogen-related application—in this case, oriented toward H_2_ purification. They constructed an MMM in situ with ZIF-8 and amidoxime-functionalized PIM-1 (AO-PIM-1), achieving remarkable performance in H_2_/CO_2_ separations, with a selectivity of 11.97 and an H_2_ permeability of up to 5688 Barrer. In addition, they demonstrated that defect-free MOF/PIM MMMs are possible when they are made on a SiO_2_ support by using an in situ growth method [139].

ZIF-8 MOF has also been used to improve the O_2_/N_2_ separation performance of PIM-1. Composite membranes were made by growing ZIF particles on the PIM-1 membrane, reaching O_2_ permeabilities of 1287 Barrer and an O_2_/N_2_ selectivity of 3.7, exceeding the Robeson upper bound line [121,128,140,141] (The Robeson Upper Bound was first defined by Robeson in 1991 as an empirical trade-off curve between gas permeability and selectivity for polymeric membranes [140]. It was updated in 2008 to reflect advances such as PIM-1 and PIM-7 [121], with further refinements in 2015 (for O_2_/N_2_, H_2_/N_2_, H_2_/CH_4_) and again in 2019 for CO_2_/CH_4_ and CO_2_/N_2_ separations [141]. Materials that surpass these boundaries (such as thermally rearranged polymers and ultra-rigid PIMs like benzotriptycene-based variants) demonstrate exceptional separation performance, often via enhanced diffusivity or dual-mode transport. Their ability to redefine the boundary sets new targets, enabling more energy-efficient and cost-effective gas separation technologies). Ma X et al. used ZIF-8/PIM-6FDA-OH MMMs to enhance propylene/propane separation, a very important industrial process. They used an ultra-microporous ZIF-8 and a dihydroxyl-functionalized polyimide (PIM-6FDA-OH), demonstrating strong interactions between the OH-functionalized polyimide and the imidazole framework between H bonding. They reported propylene permeability data significantly above the upper bound, using up to 65% ZIF content and maintaining a C_3_H_3_/C_3_H_8_ selectivity of 31 [126].

Another example is the one reported by Ma X et al., who fabricated highly propylene/propane-selective mixed-matrix membranes composed of a hydroxyl-functionalized microporous polyimide (PIM-6FDA-OH) and the well-known zeolitic imidazole framework (ZIF-8). They report an excellent compatible MMM loaded up to 65% with filler, which is a high, uncommon load for an MMM, thanks to the hydrogen bonds formed from the nitrogen and the hydroxyl groups. They reach a 3.5 Barrer permeability for the propylene and a propylene/propane selectivity of 30 [126].

However, **gas separation properties and CO_2_ capture** are the main applications of PIM/MOF-based MMMs since the high permeability of the MOFs and the adequate gas properties and processability of the PIMs allows for the creation of membranes that combine both high permeability rates with high selectivity. Moreover, the dispersibility and interface compatibility in these materials are improved by using the techniques mentioned above [48,142,143,144].

Zhang et al. reported in 2024 that CO_2_ permeability is one of the most relevant industrial applications of MMMs. They used PIM-1 combined with the CALF-20 MOF to improve the permeability, which in turn enhanced the overall gas separation performance. In the study, they used ultrasonication to reduce the particle size of the CALF-20. Only a 5% load in a 5%-CALF-20/PIM-1 membrane increased the CO_2_ permeability by 38.7% (8003 Barrer) compared with the pure PIM-1 membrane. CO_2_/N_2_ selectivity also increased up to 25%, surpassing the Robeson upper bound. However, when they increased the load to 10 or 20%, the mixed-matrix membrane did not yield improved gas separation properties, probably due to the agglomeration of the filler into the polymer matrix. They concluded that a 5% load was the limit to obtain an ultrafast CO_2_ selective membrane, thanks to the stronger attraction of the CALF-20 filler and the rigid PIM-1 molecular chains [105,145].

In another communication, Nguyen Tien-Binh et al. used a novel filler called Mg-MOF-74 (also called CPO-27) to improve the interface compatibility with PIM-1. They used specific temperature and time conditions to mix these two components (65 °C and 24 h) in chloroform, and thanks to the crosslink between the hydroxyl groups of the filler with the fluoride chain-ends of the PIM-1, the interfacial defects were minimized. The enhanced interaction allowed for the generation of interconnected 1-D hexagonal channels of 1.1 nm in diameter, forming an inter-connected micropore network that increased CO_2_ permeability by 3.2 times, from 6500 Barrer for the pure PIM-1 to 21,000 Barrer for the 20%-CPO-27/PIM-1. Meanwhile CO_2_/CH_4_ selectivity improved from 12.3 to 19.1. However, these MMMs showed a lower elasticity (a decrease of 30–70% compared with the pure PIM-1), probably due to the crosslink. They also tested the membrane after removing the MOF with acetic acid, seeing a decrease in CO_2_ permeability from 21,200 to 7500 Barrer, almost the same as that of the pure PIM-1. This experiment shows clearly the MOF contribution and the high PIM-1 stability. The also saw a decay in the CO_2_/CH_4_ and CO_2_/N_2_ MMM selectivity from 19.1 and 28.7 to 10.6 and 15.4, respectively, which did not happen in the O_2_/N_2_ and CH_4_/N_2_ ones, which were still low [146].

Another study published by Guangcan Huang et al. used the MOF filler MIL-101-HNO_3_ (also known as MIL-101(Cr), where Cr stands for chromium) due to its improved dispersibility properties in solvents and good relationship with the PIM-1 matrix due to its optimal particle size and exposure of crystallographic planes. The 15%-MIL-101-HNO_3_/PIM-1 MMMs showed an exceptional CO_2_ permeability and CO_2_/N_2_ selectivity, of 14,879 Barrer and 24.3, respectively. They used acid modulators as nitric acid to change the morphology characteristics of the MOF, resulting in reduced particle size with a better filler dispersibility and more [220] crystallographic planes, allowing the polymer to penetrate the filler crystal pores, enhancing even more the interfacial compatibility and facilitating the CO_2_ transfer inside the mixed-matrix membrane. They also performed some N_2_ adsorption experiments, with that MIL-101-HNO_3_/PIM-1 showing the minimum adsorption compared with the other MOFs under study, leading to a better selectivity versus CO_2_. In addition, they compared different permeability and selectivity results varying the filler load, and they saw that loads of 5–15% led to increased CO_2_ permeability and CO_2_/N_2_ selectivity, higher than those of PIM-1; meanwhile when increasing the filler load from 20 to 40%, the MMMs reached a CO_2_ permeability of almost 30,000 Barrer, but the selectivity decreased almost 50%. They also evidenced that the fillers did not stop polymer aging of the PIM [147].

MOF-801 was also used to improve PIM-1 CO_2_ permeability. Wenbo Chen et al. reported an MMM with enhanced CO_2_ permeability and selectivity, and this time, they improved the aging properties, with 70% of the CO_2_ permeability remaining after the 90-day aging test. Since the particle size in MOF-801 is 500 nm, the dispersion in the polymer matrix was good, which directly affected the gas separation properties. CO_2_ permeability increased from 4200 Barrer for the pure PIM-1 membrane to 9686 Barrer for the 5%-MOF-801/PIM-1 membrane. CO_2_/N_2_ selectivity also increased from 20 to 27 since PIM-1 membranes have low selectivity. The MOF cavity diameter (7.4 A) and the increased free volume in the polymer matrix led to fast and improved CO_2_ selective transport, but when the filler load surpassed 5%, selectivity started to decay and permeability increased, which indicates that the best performance was obtained with a 5% load [148]. Lower loads were used by Xuebi Du et al. and his team. They synthesized a 2%-MOF-808/PIM-1 mixed-matrix membrane that increased the CO_2_ permeability from 4022 Barrer for the pure PIM-1 to 6854 Barrer for the MMM. They also saw an increase in the CO_2_/N_2_ and CO_2_/CH_4_ selectivity of 27.4 and 1.7%, reaching a 23.2 selectivity value for CO_2_/N_2_, which is not that high compared with other MOFs [105,146,147,148] but still increased the pure PIM-1. They also tested a long-term experiment to see the gas property retention, and after 6 days they saw sustained efficacy of the PIM/MOF-808-2% membranes [42].

Another filler used not only to increase the gas permeability and selectivity but also to prevent the aging of the polymer is the MUF-15 MOF (Massey University Framework-15). Hang Yin et al. reported the first attempt to make an MMM with this MOF and PIM-1. Due to its high BET surface area (1260 m^2^/g) and pore volume (0.46 cm^3^/g), it has a very high CO_2_ adsorption capacity. It also possesses remarkable properties as a filler due to its aperture size (0.36 nm) and its crystalline structure, which allows crystalline two-dimensional nanosheets that can cover larger area of the final membrane compared with other morphologies. They saw that more than a 5% filler load led to defects at the polymer interface and agglomeration problems. Moreover, the CO_2_/N_2_ and CO_2_/CH_4_ selectivity decreased. The 15%-MUF-15/PIM-1 membranes showed a remarkable 98% increase in CO_2_ permeability compared with the pure PIM-1 but lower selectivity; thus, the 5% filler load was the most equilibrated mixture, since CO_2_ permeability increased 38% and CO_2_/N_2_ selectivity increased to 17.72. However, this result highlights lower values than for previous MMMs with other PIM-1/MOF combinations. On the other hand, they conducted 35-day aging studies and saw that 2% and 15 loads prevented aging, but 2% did so better since the 15% load had interfacial voids and filler agglomeration, which did not prevent the polymer from compacting. Furthermore, the relative CO_2_ permeability was the best preserved [149].

Yanan Wang et al. observed that combining a 20% NH_2_-ZIF-7 with PIM-1 provided MMMs with enhanced CO_2_/CH_4_ diffusion and selectivity and a strong interface, using experimental and predicted data. They used MOF nanoparticles of both NH2-ZIF-7 and ZIF-7 and PIM-1 to cast the MMMs by the solution mixing method. Thanks to the nano-particle size of the filler, they did not find any aggregation or sedimentation under a 30% filler load, but when they increased beyond this load, big aggregations and interfacial voids could be observed, leading to a deterioration of the separation and selectivity properties. NH_2_-ZIF-7 loads of 10 and 20% seemed to be good since transparent and homogeneous films were obtained. In addition, they performed gas separations tests using pure PIM-1 and MMMs. When they increased the filler load up to 20%, CO_2_ permeability decreased from 4533 Barrer for pure PIM-1 to 2953 Barrer for NH_2_-ZIF-7/PIM-1, but CO_2_/CH_4_ selectivity increased from 12.5 to 20.6. When they increased the load to 30%, the permeability increased dramatically, with the corresponding selectivity drop. In conclusion, the study reported a filler able to improve selectivity but not permeability, which makes it less interesting than other MOFs commented on before for gas separation applications. However, aging of the polymer and mechanical properties were improved. They tested ZIF-7/PIM-1 and NH_2_-ZIF-7/PIM-1 membranes and compared them with pure PIM-1. After 120 days, pure PIM lost 50% of its CO_2_ permeability, while the MMM with ZIF-7 decreased by 39% and the amino one by just 26%, concluding that aging and plasticization effects were reduced by the introduction of this NH_2_-ZIF-7 filler into PIM-1 and proving that amino groups enhance interfacial interactions, making the MMMs stronger [150].

In another study, Yang Feng et al. achieved higher values of CO_2_ permeability and even higher ones for CO_2_/CH_4_ selectivity without losing any of them, using the glass phase of ZIF-62 (agZIF-62). They used a thermal treatment that eliminates the interfacial voids between the filler and the polymer while increasing at the same time the PIM-1 free volume [137]. By heating for 30 min at 373 K and then raising the temperature to 693 K and maintaining it for 5 more minutes, they obtained agZIF-62. Then, they used the solution casting method to make two different MMMs, agZIF-62/PIM-1 and ZIF-62/PIM-1. Yang’s group used different filler loads (10%, 20%, 30%, 40%, 50%) to test the differences. When they used the ZIF-62/PIM-1 membranes, they found an increase in the CO_2_/CH_4_ selectivity with respect to that of the PIM-1 pure membrane to 37.2, 41.5, 49.8, 39.6, and 30.8, respectively. Moreover, when they used a 30% filler load, they found a maximum increase in selectivity. In addition, the maximum CO_2_ permeability was found for the 50% filler load. However, when the experiment was repeated with the homologous membrane with agZIF-62/PIM-1 in the same ratios, they found a similar increase for the CO_2_ permeability, a decrease in the CH_4_ permeability in proportion to the filler load, and a marked increase in the CO_2_/CH_4_ selectivity to up to 67 using a 30% load. For this MMM ratio they also found a CO_2_ permeability of 5914 Barrer, much higher than the values corresponding to PIM-1 pure membrane (4654 and 18.1). In both cases, they surpassed the Robeson upper bound line of 2019 and most of the state-of-art MOF-based MMM CO_2_/CH_4_ selectivity [151].

On the other hand, Chumei Ye and his team incorporated a nano-sized ZIF-67 (ZIF-S) and a submicro-sized ZIF-67 (ZIF-L) with larger surface areas and better compatibility to PIMs to enhance the selectivity. Different loads of filler, from 5 to 20%, were compared. In all cases, they saw by XRD that when increasing the filler load, membrane density increased, giving rise to a reduced fractional free volume, which is in accordance with the variation in the d-spacing value observed, which verifies that polymer chain rigidization occurred that was induced by the well interfacial interactions with the filler. When they performed separation tests for the ZIF-S/PIM-1 membrane, they found a decrease in CO_2_ permeability with an increase in filler load from 4521 for the pure PIM to 2805 for the 15% load MMMs, with the permeability decrease being slightly lower for the 20% filler content. The maximum CO_2_/CH_4_ selectivity was found for a 15% ZIF-S load, increasing from 12.45 to 21.09, which is 69.4% higher. All the other selectivity also increased (CO_2_/N_2_: 36.2%, H_2_/CH_4_: 99.7%, and H_2_/N_2_: 60.7%). Surprisingly, when they repeated the same experiment using ZIF-L as the load, the permeability increased with the filler content, reaching a maximum for 20%, but both selectivity values decreased compared with the ZIF-S load. This is well known since ZIF-8 performs very similarly [128]. Good interfacial compatibility of ZIF-S particles with the polymer matrix could facilitate the formation of partial pore blockage regions, generating improvements in CO_2_/CH_4_ selectivity but resulting in lower CO_2_ permeability due to the narrowed effective aperture size of ZIF-S particles. When the filler content was increased to 20% in the ZIF-S/PIM membrane, the CO_2_ permeability increased, which can be attributed to the interfacial defects leading to non-selective gas diffusion, which was also proven by the aggregation observed by SEM.

Finally, they performed a CO_2_ permeability retention test measured after 120 days and observed a 70% permeability maintenance, compared with 50% for the pure PIM-1 membrane [82].

The data collected in Table 2 demonstrates that the performance of MOF/PIM-based MMMs in gas separation applications is highly dependent on both the choice of MOF filler and the targeted gas pair. For hydrogen separations, the CAU-12-ODB/PIM-1 membrane stands out for its outstanding permeability and exceptional H_2_/N_2_ selectivity. In the case of O_2_/N_2_ separation, although only one example (ZIF-8/PIM-1) is reported, its performance appears promising for further development. For propylene/propane separations, the ZIF-8/PIM-6FDA-OH MMM provides favorable selectivity, even if its permeability is modest. Regarding CO_2_ separation, a trade-off between permeability and selectivity is observed. On one hand, the Mg-MOF-74/PIM-1 membrane exhibits the highest CO_2_ permeability, but with only moderate selectivity. On the other hand, MIL-101-HNO_3_/PIM-1 offers an excellent balance for CO_2_/N_2_ separations, combining high permeability with respectable selectivity. Additionally, for CO_2_/CH_4_ separation, the agZIF-62/PIM-1 membrane emerges as the best option with the highest selectivity, even though it may not reach the peak permeability values observed in other cases.

Overall, all these data indicate that by carefully selecting the type of MOF filler and optimizing its loading within the PIM matrix, it is possible to tailor the membrane’s properties to meet specific gas separation requirements.

## 4. MOF-COF Hybrid Materials

The development of hybrid materials based on MOFs and COFs has emerged as a rapidly growing field of materials research. First described in the late 1990s, **metal–organic frameworks (MOFs)** are crystalline materials composed of metal ions or clusters coordinated to organic linkers, resulting in extremely porous structures [153,154]. MOFs have received a lot of interest for their large surface areas, variable porosity, and functional diversity in applications including gas storage, catalysis, and sensing [155,156,157,158,159]. However, their resistance to strong chemical or heat conditions remains a significant constraint in several circumstances [160,161].

**Covalent organic frameworks (COFs),** introduced in the mid-2000s, are composed entirely of light elements (e.g., carbon, hydrogen, nitrogen, oxygen) connected through covalent bonds. Their highly ordered, crystalline nature and permanent porosity make them suitable for applications requiring chemical robustness and long-term stability [162,163,164,165]. COFs also exhibit chemical versatility and an ease of functionalization comparable to MOFs. Despite these advantages, COFs often lack the tunable electronic and catalytic properties provided by metal centers like those found in MOFs [166].

The **combination of MOFs and COFs** benefits from the complementary strengths of both materials, with the high structural diversity and tunable functionalities of MOFs being combined with the chemical stability and well-defined pore architectures of COFs [167]. This allows the limitations inherent to each class of material to be overcome, leading to the design of multifunctional materials with improved performance in various fields [24,168]. As a case in point, the enhanced thermal and chemical stability, greater structural diversity, and optimized porosity offered by MOF–COF hybrids make them a highly promising material for advanced applications such as catalysis, energy storage, and gas separation. The following table (Table 3) presents a comparative overview of the characteristics of MOFs, COFs, and their hybrid materials.

### 4.1. Type of MOF/COF Hybrids

There are several types of MOF/COF hybrids based on how the MOF and COF components interact with each other.

#### 4.1.1. Strategies for the Synthesis of MOF-COF Hybrid Materials

The combination of metal–organic frameworks (MOFs) and covalent organic frameworks (COFs) into hybrid materials has garnered significant attention due to their complementary properties. Different synthesis strategies have been developed to fabricate MOF-COF hybrids, including the called MOF-first strategy, the COF-first strategy, and post-synthetic linking of pre-synthesized MOFs and COFs (Figure 4) [169]. Each approach offers distinct advantages in terms of structural stability, functional integration, and synthetic control.

#### 4.1.2. MOF-First Strategy

In this approach, the MOF is synthesized first and subsequently serves as a template for COF growth, resulting in MOF@COF hybrid materials. This method enhances the stability and functionality of the final composite by leveraging the inherent porosity and structural integrity of the MOF (Figure 4). Several variations of this strategy have been explored [170,171,172].

One commonly employed method is in situ COF growth, in which the COF is directly formed on the MOF surface. This process can involve either covalent bonds, which provide a strong and stable interaction, or non-covalent interactions, which offer a more flexible but weaker linkage. A typical example of this strategy is the development of core–shell structures, where the MOF forms the core and the COF forms the shell, enhancing the material’s surface area and catalytic activity [106,173].

Another variation involves the transformation of an amorphous layer into a crystalline COF layer. Initially, a non-crystalline layer of COF precursors is deposited on the MOF surface. Through thermal or chemical treatments, this layer undergoes structural rearrangement, leading to the formation of a well-defined crystalline COF. This method ensures better integration between the two frameworks and results in a highly ordered hybrid material [174,175,176].

A third approach under the MOF-first strategy is one-step COF synthesis, where the MOF and COF are synthesized simultaneously within a single reaction process. This simplifies the fabrication method and promotes direct growth of the COF onto the MOF. However, achieving uniform structures requires precise control over reaction parameters to balance the nucleation and growth kinetics of both components [177,178,179].

#### 4.1.3. COF-First Strategy

In contrast to the MOF-first strategy, the COF-first strategy involves the initial synthesis of the COF, which subsequently acts as a scaffold for MOF growth (Figure 4). This method is particularly useful when the COF’s intrinsic porosity and functional groups need to be pre-established before the MOF is introduced. Two main techniques have been developed within this framework.

The layer-by-layer growth approach involves the sequential deposition of MOF layers onto the COF surface. This method enables fine control over the thickness and composition of the MOF layer, leading to tunable hybrid structures with tailored properties. The gradual growth process ensures homogeneous coverage of the COF and facilitates the formation of well-defined MOF domains.

Another technique is sequential MOF-on-COF growth, in which the MOF is grown on the COF surface through controlled chemical reactions. This approach ensures strong adhesion between the two frameworks, leading to a robust hybrid material with enhanced mechanical stability and chemical resilience [180,181,182].

#### 4.1.4. Post-Synthetic Linking of Pre-Synthesized COFs and MOFs

A third strategy for synthesizing MOF-COF hybrid materials involves the post-synthetic linking of separately prepared MOFs and COFs (Figure 4). In this approach, both components are synthesized independently and then integrated through either covalent bonding or physical interactions. Covalent bonding offers a strong and stable linkage between the MOF and the COF, facilitating the formation of durable hybrid materials with well-defined interfaces. Alternatively, physical interactions such as hydrogen bonding or π–π stacking provide a more flexible means of integration, allowing for dynamic hybrid structures that can be tuned for specific applications. This post-synthetic method provides high versatility, as different MOF-COF combinations can be designed and assembled based on desired properties and functionalities [170,171,172].

The selection of the appropriate synthetic strategy for MOF-COF hybrids depends on the desired outcome for the material’s stability, structural integrity, and functionality. The MOF-first approach is best for enhancing the stability and performance of MOFs. The COF-first strategy is ideal for structural control and design flexibility, and post-synthetic linking allows for modular, customizable assembly of hybrid materials (Table 4).

### 4.2. Applications of MOF/COF Hybrids

Hybrid materials derived from the integration of metal–organic frameworks (MOFs) and covalent organic frameworks (COFs) have emerged as promising candidates for a wide range of advanced applications. Owing to their complementary structural and chemical features, these hybrid systems exhibit enhanced performance in various fields. The following sections provide an overview of their potential in key application areas, including pollutant adsorption, catalysis, energy storage and conversion, sensing technologies, biomedical applications, and gas adsorption and separation (Figure 5) [175,177,182,188,189,190].

#### 4.2.1. Sensors

The combination of metal–organic frameworks (MOFs) and covalent organic frameworks (COFs) has led to the development of advanced sensor materials with enhanced sensitivity and selectivity. For instance, a hybrid material composed of Ce-MOF@MCA was synthesized by combining a melamine–cyanuric acid COF with a cerium-based MOF. This hybrid demonstrated high electrochemical activity and strong bio-affinity, leading to the development of an aptasensor for oxytetracycline (OTC) detection with an extremely low detection limit of 35.0 fM [182]. Another example is the Co-MOF@TPN-COF-based aptasensor, which was used for detecting ampicillin, showing a remarkable limit of detection (LOD) of 0.217 fg/mL [181]. Furthermore, a UiO-66-NH2@COF hybrid was employed in electrochemical analytical methods for adenosine triphosphate (ATP) and chloramphenicol (CAP) detection, achieving a detection limit of 5.04 fg/mL [191]. Another hybrid with the same MOF, UiO-66-NH2@TpTt-COF (UNT), a core–shell MOF-COF composite, is used for tetracycline detection due to its dual emissions and its ability to suppress aggregation-caused quenching (ACQ), resulting in high sensitivity and specificity. UNT is used for the detection of tetracycline in soil and river water, with recoveries of 92.96–98.44% and 88.58–104.31%, respectively [192].

#### 4.2.2. Catalysis

MOF–COF hybrids have demonstrated remarkable potential in catalytic applications, particularly in heterogeneous and photocatalysis applications. They are perfect candidates for catalytic reactions because of their structural characteristics, which include stability, well-defined active sites, and large surface areas.

The catalytic and photocatalytic applications of MOF-COF hybrids have been extensively explored, showcasing their exceptional potential in various reactions. A prime example is the PCN-222-Co@TpPa-1 hybrid, where a covalent organic framework (COF), TpPa-1, is grown on a cobalt-based metal–organic framework (MOF), PCN-222-Co. This hybrid demonstrated remarkable recyclability and high catalytic activity in the deacetalization–Knoevenagel reaction [193]. Similarly, the sandwiched Pd/UiO-66-NH_2_@COF composite exhibited selective catalytic activity in hydrogenation reactions, driven by the molecular size, offering a tailored approach to catalytic processes [175].

In the field of photocatalysis, MOF-COF hybrids have also garnered significant attention. The NH_2_-UiO-66@TFPT-DETH hybrid achieved a hydrogen evolution rate of 7178 μmol/g/h, while another hybrid, NH_2_-UiO-66/TpPa-1-COF, demonstrated an even superior photocatalytic hydrogen evolution rate of 23.41 mmol/g/h [168]. These materials highlight the impressive photocatalytic efficiency of MOF-COF hybrids. In addition to these materials, the IR-MOF3@COF-LZU1 hybrid exhibited high photocatalytic activity for the degradation of p-nitrophenol [194].

Another study demonstrated the efficiency of MOF/COF hybrids in wastewater remediation, where methylene blue pollutants were effectively degraded using visible light and hydrogen peroxide as co-reactants. NH2-MOF-5/MCOF hybrid showed high visible light absorption (up to 596 nm) and a reduced band gap (2.20 eV), facilitating the photodegradation of dyes such as methylene blue (69% degradation in 5 h), which significantly improved with the addition of H_2_O_2_ (degradation rate of 0.728 h^−1^) [195]. These results indicate their utility for environmental applications. Moreover, the integration of metal nodes in MOFs with nitrogen-rich COF frameworks has been employed to develop materials capable of catalyzing CO_2_ reduction reactions, showcasing their potential in addressing environmental challenges related to greenhouse gas emissions. The PCN-222-Cu@TpPa-1 (1:2) hybrid showed a CO yield of 90.57 μmol g^−1^h^−1^ and a CH_4_ yield of 21.27 μmol g^−1^h^−1^ in the photocatalytic reduction of CO_2_, outperforming other materials. CH_4_ selectivity could be improved by adjusting the ratio of the MOF and COF. This material had a specific surface area of 577.8 m^2^g^−1^, demonstrating high catalytic efficiency. In addition, the stability of the material was observed after eight cycles, with minimal variations in its structure [196].

These findings underscore the versatility and high performance of MOF-COF hybrids in both catalytic and photocatalytic applications.

#### 4.2.3. Gas Sorption and Separation

Gas sorption and separation are among the most well-researched applications of MOF–COF hybrids due to their tunable porosity and high adsorption capacities. Recent advancements have demonstrated their use in CO_2_ capture, hydrogen storage, and natural gas purification.

The use of MOF-COF hybrids for gas storage and separation has been investigated due to their high surface areas and tunable porosities. A UiO-66-NH_2_@COF-TAPB-BTCA composite displayed enhanced N_2_ and H_2_O uptake, making it suitable for water sorption applications [197]. Similarly, COF-300 was grown on UiO-66 membranes to produce a [COF-300]-[UiO-66] composite membrane with high gas selectivity [198]. Another hybrid membrane, [COF-300]-[Zn_2_(bdc)_2_(dabco)], showed improved separation selectivity for H_2_/CO_2_ mixtures [180].

Pyridine-based COFs grown on NH_2_-UiO-66 cores showed large specific surface areas and favorable chemical stability, exhibiting significantly enhanced CO_2_ adsorption, with MOF@COF-0.05 reaching an uptake of 4.53 mmol⋅g^−1^. Functionalization with bromoethanol created MOF@COF-0.15-Br, which facilitated efficient catalytic conversion of CO_2_ to cyclic carbonates with high yields. This functionalized catalyst demonstrated stability, with negligible loss of activity after six cycles. Compared to other materials, MOF@COF-0.15-Br did not require co-catalysts, metals, or solvents for CO_2_ conversion [199]. Regarding H_2_ adsorption, the hybrid material MIL-101(Cr)@UiO-66(Zr) showed a higher H_2_ adsorption capacity than the base materials (MIL-101 and UiO-66). This material showed an increase in adsorption of up to 26% compared to MIL-101 and 60% more than UiO-66 [200].

#### 4.2.4. Adsorption and Removal of Pollutants

MOF-COF hybrids have demonstrated excellent adsorption capabilities for environmental remediation. Mn-1,4-BDC/COF (Mn-1,4-benzenedicarboxylate MOF combined with an imine-linked melamine and terephthalaldehyde COF) composite is a robust adsorbent for the rapid and simultaneous removal of auramine O (AO) and rhodamine B (RB) dyes from aqueous media. It exhibited superior adsorption capacity (25.13 mg/g for AO and 21.53 mg/g for RB) compared to individual MOF Mn-1,4-BDC and COF. This improvement was due to synergistic effects that allow the simultaneous use of multiple adsorption mechanisms [177]. NH_2_-MIL-125(Ti)@TpPa-1 hybrid exhibited high adsorption capacity for uranium (536.73 mg/g) and europium (593.97 mg/g) [201]. Additionally, a Zn-MOF-5 and melamine–terephthaldehyde COF hybrid showed outstanding performance in removing organic pollutants from wastewater [168].

#### 4.2.5. Energy Storage and Conversion

The high surface area, tunable pore sizes, and electrical conductivity of MOF–COF hybrids make them excellent candidates for energy storage and conversion applications, including batteries, supercapacitors, and electrocatalysis.

The incorporation of MOFs into COFs enhances the electrochemical performance of supercapacitors. For instance, a novel MOF@COF-TCNQ material demonstrated a high area capacitance of 78.36 mF cm^−2^ and a volumetric energy density of 4.46 F cm^−3^, thanks to TCNQ infiltration increasing the quantum capacitance, with MOF being UiO-66-NH2 and COF being TpPa [202]. For electrodes in supercapacitors, the UiO-66-NH_2_@TFP-TABQ hybrid combines the structural advantages of both MOFs and COFs. Although the MOF possesses the largest surface area (478 m^2^ g^−1^), the hybrid exhibits superior porosity, with an area of 236 m^2^ g^−1^, six times larger than that of the COF (39 m^2^ g^−1^). In addition, it exhibits outstanding pseudocapacitive behavior, with a b-value of 0.85 (compared to 0.59 for MOF and 0.83 for COF), indicating a higher contribution of redox reactions in its charge storage capacity [203]. Another aza-MOF@COF composite, synthesized via an aza-Diels–Alder cycloaddition, combines the porosity of MOF (UiO-66-NH_2_) with the stability of COF (LZU-1) and is further enhanced by quinoline modification. This hybrid material demonstrated a high specific capacitance of 20.35 mF/cm^2^ and a volumetric energy density of 1.16 F/cm^3^, making it highly effective for supercapacitor applications. It also exhibited excellent cycling stability, retaining 89.3% of its capacitance after 2000 cycles. The quinoline modification plays a key role in improving energy storage performance and structural robustness [204].

An interlinked hybrid of imine-based COFs and Mn-based MOFs (COF/Mn-MOF) demonstrated superior lithium storage performance, with a specific capacity of 1015 mAh/g after 650 cycles [205].

#### 4.2.6. Biomedical Applications

A series of MOF@COF hybrid materials have been developed for biomedical and electrochemical applications, combining the properties of metal–organic frameworks (MOFs) and covalent organic frameworks (COFs). The NMCTP-TTA nanozyme integrates a peroxidase-like MOF (NH2-MIL-88B(Fe)) with a COF (COFTP-TTA) to enhance bacterial inhibition. The MOF serves as the catalytic center, while the COF’s hierarchical nanocavities activate substrates, mimicking natural enzymes. Pseudopodia-like structures on the COF facilitate bacterial capture. NMCTP-TTA generates reactive oxygen species (ROS), exhibiting a 7.9-fold increase in peroxidase-like activity over NM-88. Functional groups such as triazine and phenol boost proton and electron transfer, significantly improving bacterial inhibition and wound healing outcomes [190]. A Cu-MOF@TpBD hybrid material has been designed for electrochemical detection of PDGF-BB in human serum. The Cu-MOF core generates electrochemical signals without mediators, while the COF shell stabilizes the sensor and enables aptamer immobilization via π–π stacking and hydrogen bonding. This combination enhances bioanalyte sensitivity and signal stability. The aptasensor demonstrated excellent recovery rates (91.0–107.9%) and low RSD values (2.9–4.5%) in human serum samples. The hybrid material provides a highly selective and reproducible platform for PDGF-BB detection [206]. Targeting GL-3 antigen, a Ti-MOF@COF composite has been developed for a sandwich-type immunosensor. The hybrid material enhances biomolecule compatibility and provides a stable antibody immobilization framework. Its large surface area (1403.3 m^2^ g^−1^) amplifies signal detection, resulting in an ultra-sensitive response. The immunosensor achieves a linear range of 0.0001–20.0 ng mL^−1^ with a sensitivity of 0.025 pg mL^−1^. Ti-MOF-modified glassy carbon electrodes (GCE) present the highest amperometric response, confirming the material’s superior electrochemical performance [207]. A Cu-MOF@CuPc-TA-COF hybrid material has been designed for dual photoelectrochemical (PEC) and electrochemical (EC) detection of HIV-1 DNA. The Cu-MOF core enhances electrochemical and photoactivity, while the COF framework improves stability and probe immobilization. The hybrid structure provides a high BET surface area (514.7 m^2^·g^−1^) for efficient DNA detection. Detection limits reach 0.07 fM (PEC) and 0.18 fM (EC) within a 1 fM–1 nM linear range. The biosensor demonstrates high accuracy, with recovery rates of 90.2–114.8% and low RSD values (0.2–2.3%) in human serum samples [208].

## 5. MOF/HCP, MOF/CTF, and MOF/CMP Hybrid Materials

As already discussed in the first part of the review, due to the results in different research fields [209,210,211,212,213], MOFs have been hybridized with other materials [24,48,214] in order to improve their properties. Among these materials are porous materials—more specifically, POPs, which have recently been postulated as a promising alternative to improve the properties of MOFs [215]. PIMs [110] and COFs [216] are the most widely used compounds to synthesize hybrid compounds with MOFs and offer great results in different applications, such as catalysis, gas storage, and molecular separation [215]. However, relatively recently, crosslinker polymers [25,28], covalent triazine frameworks [33,34,217], and hyper-crosslinker polymers [218]—another set of POPs with very good properties for different applications that have received considerably less attention because they have not been studied as thoroughly as the other POPs, but have great potential for hybridization with MOFs—have begun to be used [215]. The hybridization of the MOFs CMP, CTF, and HCP allows the structural and functional advantages of these porous materials to be combined with the inherent properties of MOFs.

MOF/HCP hybrids:

The hybridization of MOFs with HCPs brings together the tunable porosity and chemical characteristics of MOFs with the thermal stability and versatility of HCPs [31]. These hybrids are particularly promising in gas capture and catalysis applications [219,220,221]. In addition, the ease of preparation of HCPs and their compatibility with various chemical conditions broaden the range of possible applications for these materials [219,222].

2.MOF/CTF hybrids:

Hybrids of MOFs with CTFs take advantage of their structural rigidity and ability to handle extreme conditions, making them interesting materials for catalytic and gas separation applications [33,217]. The thermal and chemical stability of CTFs, coupled with the tunable functionality and high porosity of MOFs, make them particularly attractive for catalytic and gas separation applications [36,38]. However, their synthesis and the control of the MOF-CFT interfaces are areas of current active research.

3.MOF/CMP hybrids:

MOFs can also be combined with CMPs to create hybrid materials that take advantage of their π-conjugated systems to allow for good electronic conductivity, making them ideal for applications related to photocatalytic catalysis and gas storage [27,223,224]. Such hybrids have shown potential in energy-related processes such as hydrogen production by photocatalysis, as well as in electronics applications [26,29].

As discussed above, HPC/MOF, CMP/MOF, and CTF/MOF systems have been less studied, despite offering complementary properties such as greater structural diversity, electronic tunability, and chemical stability. Below, we highlight the most relevant research on the hybrids between these compounds with a diversity of applications, with the aim of improving the individual properties of each one and overcoming the limitations of each of the separate components.

One of the first studies on these POPs hybridized with MOFs to improve their properties was carried out in 2013 when they used a hyper-crosslinked polymer to encapsulate MOFs in the micropores of the polymer to store hydrogen. They used hyper-crosslinked styrene–maleic acid beads decorated with a MOF to synthesize SMA-MOF-5, SMA-Zn-BPDC, and SMA-Zn-NDC. The last hybrid composite performed the highest, with a storage capacity of 0.61 wt% at 300 K and 0.69 bar for 24 h [219]. One of the applications where both POPs and MOFs are most used is in the adsorption of compounds, thanks to their structure, which is why researchers have recently been trying to find synergies between the two to improve their properties, as mentioned above. Li S et al. succeeded in introducing MOF MIL-101(Fe)-NH_2_ into a hyper-crosslinked IRA-900 resin to improve its phosphonate adsorption properties by enhancing the properties of each component separately. Separately, MIL-101(Fe)-NH_2_’s adsorption capacity is 7.6 mg P/g in 240 min, IRA-900’s adsorption capacity is 5.6 mg in 360 min, and the MOF/HPC hybrid has a capacity of 12.9 mg in 120 min, in addition to a high selectivity to other anions [220]. On the other hand, MIL-101(Cr) together with hyper-crosslinked aniline polymer was also used for the separation of Cr(VI), forming a series of compounds called HPAN@M-x. Specifically, HPAN@M-b (b = 5 mmol aniline) achieved a Cr(VI) adsorption of 290.0 mg/g in 7 min and could be transformed into BaCrO_4_ for use, and very good stability was observed, maintaining the Cr(VI) removal during five cycles [222].

Another approach for improving the properties of these hybrid composites is the introduction of carbon or the preparation of carbon from microporous solids as precursors, such as HPCs. For example, the use of graphene oxide together with MOFs has been extensive, improving selectivity, resistance to acids and bases, and basically modifying the pore structure, among other characteristics, thus allowing for its use in a wide variety of applications [225,226]. For example, for CO_2_ separation, Ning H et al. succeeded in synthesizing a new hyper-crosslinked polyimide with UiO together with graphene oxide (PI-UiO/GO-1), which increased the amount of CO_2_ adsorbed by three times and four times compared to the CO_2_ selectivity against the nitrogen in UiO-NH_2_ MOF. The CO_2_ adsorption of the hybrid described in the study was 8.24 mmol/g at 298 K and 30 bar versus 2.8 mmol/g for UiO66-NH_2_. On the other hand, the PI-UiO/GO-1 hybrid showed a selectivity of 64.71 vs. 15.43 CO_2_/N_2_ for the MOF UiO-66-NH_2_ [227]. From a different approach, Bauza et al. synthesized a magnetic hybrid carbon MOF from a benzene hyper-crosslinked polymer together with MIL-100(Fe), with diclofenac adsorption properties of 210 mg/g for diclofenac adsorption due to improved surface area and pore interactions [32].

These compounds can also be used as catalysts for reactions that, as in other cases, obtain the advantages of MOFs on the one hand and porous polymers on the other hand. For example, Karimi et al. synthesized PC4RA@PrSi-CA/Cu-MOF. This compound substantially reduces side reactions by directing the use of organic solvents, and it can be reused up to 10 times, improving the yield of organic reactions substantially [221]. Another very current topic is the self-healing of polymers, especially in the field of elastomers, as the increasing demand for this type of compound has led to an increase in the maximum lifetime of these compounds. In the work carried out by R. Wang et al., they synthesized hybrid membranes using a hyper-crosslinked hyperbranched polymer such as HPMA together with the MOF UIO-66-NH_2_, diethylenetriamine, and terephthaldehyde, forming a so-called hyper-crosslinked hyperbranched polymer @ metal–organic polyhedral (HHMOP). These membranes achieved 91.35% strength recovered after 24 h, extending the use of MOFs and polymeric membranes [228].

CTFs are relatively recently synthesized compounds, but have great potential in different applications. This is why the synthesis of CTF/MOF hybrids is being pursued for the improvement of the separate materials. The evolution of the catalysis of hydrogen is crucial for sustainable hydrogen production, energy storage, and other reasons, which is why much research focuses on it. For example, Zr-UiO-66-NH_2_ (UN) is a MOF with good photocatalytic capabilities for H_2_ evolution. In 2023, Dong, S. et al. used this compound together with benzoic acid-modified covalent triazine-based frameworks. The photocatalytic H2 ratio was 378 µ/h-g for 30NUBC, a much higher result than with the MOF and the CTF alone [35]. Shao et al. were looking for a new compound to improve the catalytic properties for this reaction, and they synthesized a series of compounds using Ni-CAT, a conductive MOF, together with CTF-1, a photocatalyst widely used individually. At a 1:19 ratio, Ni-CAT-1/CTF-1 increased the photocatalytic activity, with a 8.03 mmol/h-g H_2_ evolution rate versus 1.96 mmol/h-g for CTF-1 [36]. This demonstrates once again the potential of hybrid materials, especially in the field of catalysis. With greenhouse gases such as SO_2_ and NO_2_ increasing on the planet, solutions are being sought in different ways, and one with great potential is the use of MOFs and POPs, which, thanks to their structure, offer promising results. For this purpose, Sumin Li et al. used CTF to synthesize Zr-MOF-NH_2_ in situ by adding CU^2+^ to obtain more active centers, obtaining the porous hybrid Zr-MOF-NH_2_/CTF-CU^2+^. The SO_2_ capture was 39.3 mg/g, which was 2.6 times more than the SO_2_ capture with the MOF, while the NO_2_ capture was 27.3 mg/g, which was 3.1 times more than with the pristine MOF [38].

Crosslinker polymers are a group of POPs that are booming because of their incredible surface area diversity, building blocks permanently, which has led to the use of these composites for a wide variety of applications [25,28,229]. Hybrids of CMP and MOFs have great potential for the synergistic properties that they can provide. There is very little research that focuses on these hybrids, as hybrids with PIMs and COFs are of much greater importance. Catalysis, as has been observed in other hybrids, is a very important topic nowadays. For example, Bo Chen et al. managed to synthesize the hybrid NU/NUF/FNBZ. NU is a MOF with a good CO_2_ adsorption capacity, and FNNZ is a CMP with visible light absorption capacity whose synergistic properties allow for an increase in its catalytic activity. The 40NU/FNBZ composite showed a CO formation rate of 18.78 µmol/h-g, four times higher than that of FNBZ and 48 times higher than that of NU [230].

## 6. Perspectives on MOF/PIM, MOF/COF, MOF/HCP, MOF/CTF, and MOF/CMP Hybrids

While MOF/PIM and MOF/COF hybrids have demonstrated superior development and a wider variety of applications, hybrids based on HCPs, CTFs, and CMPs are distinguished by their unique properties, which position them as alternatives with significant potential. These materials possess the capacity to overcome specific challenges, such as demonstrating superior thermal stability in extreme situations or exhibiting enhanced electric conductivity. Even though research on MOF/HCP, MOF/CTF, and MOF/CMP hybrids is in its early stages, their complementary characteristics indicate significant potential for future technological applications. As synthesis and characterization methods are developed and improved, these systems are expected to become increasingly influential in fields such as catalysis, energy storage, and gas separation.

Looking to the future, the field of MOF-based hybrids shows tremendous promise. MOF/PIM and MOF/COF hybrids, for example, benefit from relatively mature synthetic methodologies and have already demonstrated remarkable performance in applications such as selective gas separation, membrane technologies, and advanced catalysis. Their processability and tunable porosity also make them ideal candidates for integration into scalable technologies.

However, hybrids with HCPs, CTFs, and CMPs offer complementary benefits, including exceptional thermal and chemical stability, higher electric conductivity, and the potential for tailored electric or catalytic functions. These features are of relevance in the context of challenging applications, including environments characterized by aggressive catalysis, electrochemistry, and next-generation energy storage systems.

Despite these promising perspectives, several challenges must be addressed to fully unlock the potential of MOF-based hybrid materials. Achieving precise control over hybrid architectures, improving compatibility between MOF and polymeric phases, and ensuring reproducibility and scalability are critical. Furthermore, a more comprehensive understanding of the molecular-level interactions that give rise to synergistic properties will be essential to guide rational design. The real-world applicability of these materials will be determined by several factors, including environmental considerations. Such considerations may take the form of sustainable synthesis routes and recyclability.

In conclusion, the continuous evolution of MOF-based hybrids incorporating PIMs, COFs, HCPs, CTFs, and CMPs presents a significant opportunity to extend the scope of porous materials beyond their current limitations. The combination of the structural tunability and high porosity of metal–organic frameworks (MOFs) with the diverse functionalities and robustness of organic porous networks has the potential to drive innovation across multiple sectors. Advances in synthetic techniques, characterization methods, and fundamental understanding have led researchers to believe that these materials have the potential to become key enablers of future technologies in the fields of catalysis, energy storage, and separation science.

## Figures and Tables

**Figure 1 polymers-17-01941-f001:**
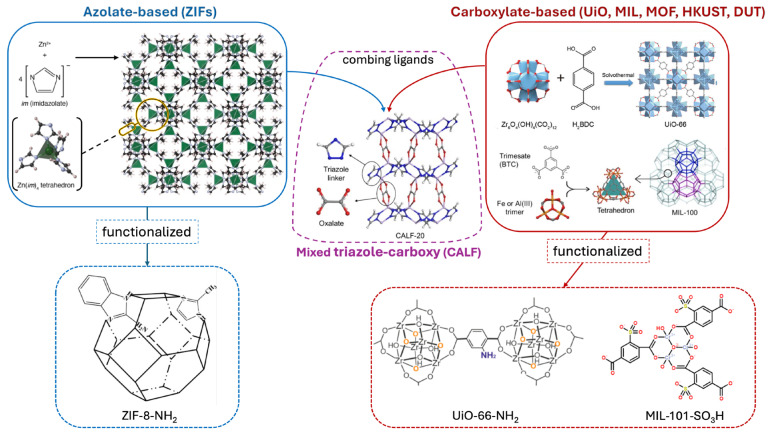
Schematic representation of the two most common types of metal–organic frameworks based on the nature of the ligand, whether it is azolate-based (ZIFs) or carboxylate-based (UiO, MIL, MOF, HKUST and DUT), and modification by combining ligands or by functionalization of their ligands to later enhance MOF–polymer interaction. Adapted with permission from Ref. [53] copyright American Chemical Society 2025, [79] copyright Elsevier 2021, [89] copyright Wiley-VCH 2023, [90] copyright MDPI 2023, [94] copyright American Chemical Society 2022, and [105] copyright Royal Society of Chemistry 2025.

**Figure 2 polymers-17-01941-f002:**
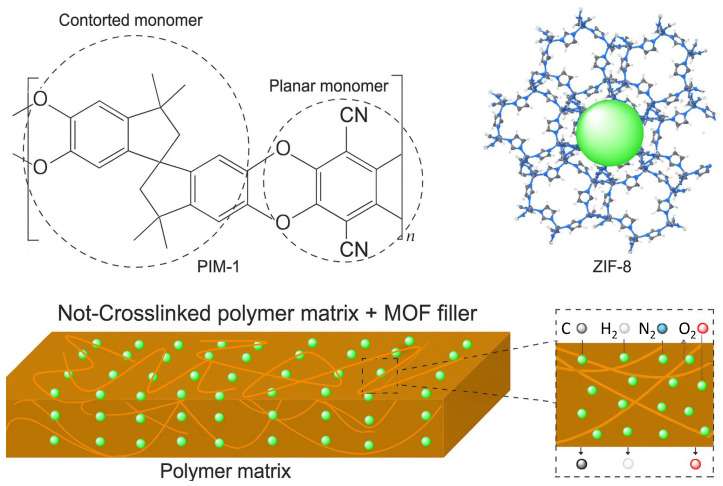
Schematic representation of a hybrid membrane composed of a PIM polymer matrix (shown in orange) and dispersed MOF particles (shown in green). This integration combines the high intrinsic microporosity of PIMs with the crystalline structure of MOFs, mitigating the individual limitations while enhancing performance in applications like gas separation, CO_2_ adsorption, and catalysis.

**Figure 3 polymers-17-01941-f003:**
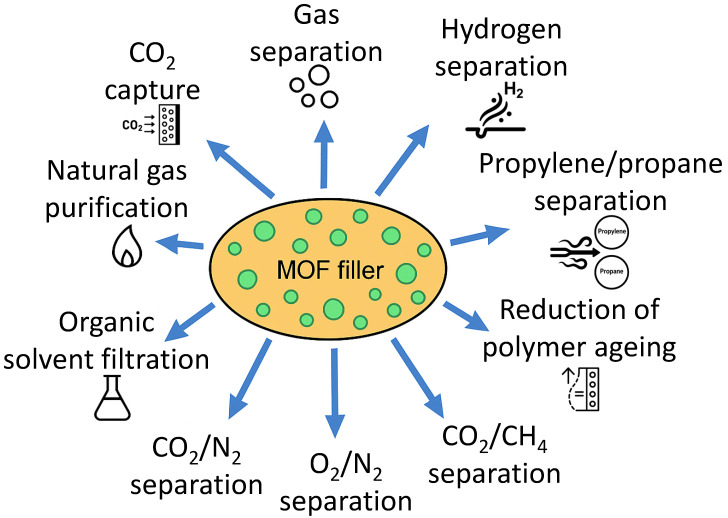
Main applications of MOF/PIM MMMs in different fields.

**Figure 4 polymers-17-01941-f004:**
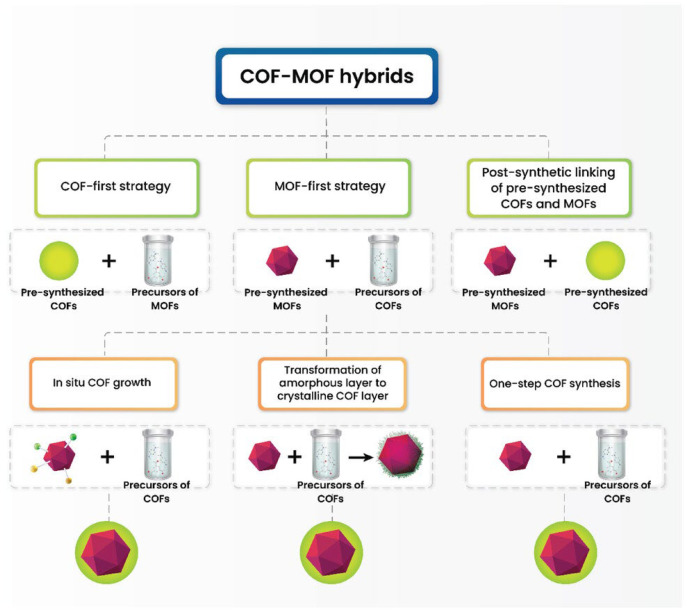
Schematic illustrating the synthetic strategies of COF–MOF hybrids. Reproduced under the terms of the Creative Commons CC-BY 4.0 license (https://creativecommons.org/licenses/by/4.0/) [169]. Copyright 2023 the authors. Advanced Functional Materials published by Wiley-VCH GmbH.

**Figure 5 polymers-17-01941-f005:**
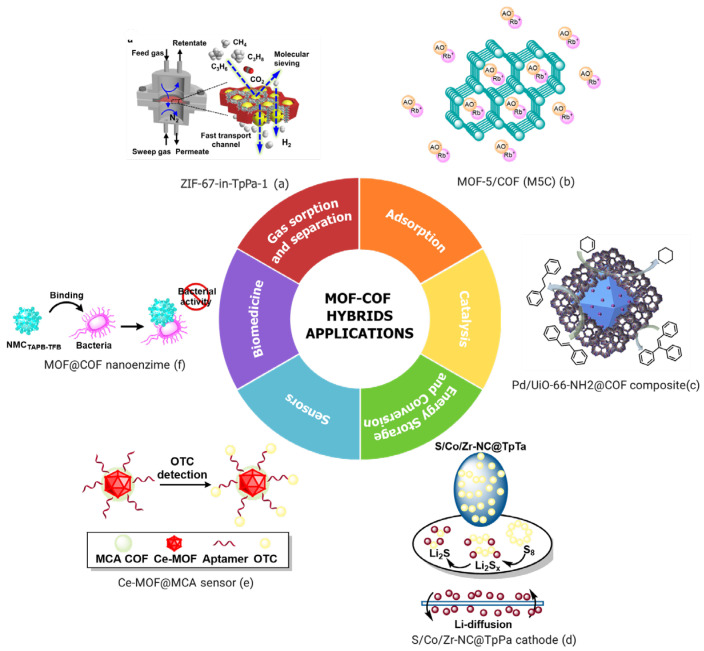
Applications of MOF-COF hybrids. (**a**) ZIF-67-in-TpPa-1 membrane for hydrogen separation. Reproduced under the terms of the Creative Commons CC-BY 4.0 license (https://creativecommons.org/licenses/by/4.0/) [188]. Copyright 2021 the authors. Nature Communications published by Springer Nature. (**b**) MOF-5/COF (M5C) for removal of auramine O and rhodamine B [177]. (**c**) Pd/UiO-66-NH2@COF composite for olefin hydrogenation catalysis. Reproduced under the terms of the Creative Commons BY-NC-ND 4.0 license (https://creativecommons.org/licenses/by-nc-nd/4.0/) [175]. Copyright 2020 the authors. Cell Reports Physical Science published by Elsevier. (**d**) S/Co/Zr-NC@TpPa cathode for Li-S battery [189]. (**e**) Ce-MOF@MCA sensor for OTC [182]. (**f**) MOF@COF nanoenzime with bacterial inhibition abilities [190].

**Table 1 polymers-17-01941-t001:** Most used pristine MOF types and their surface area, main characteristics, advantages and disadvantages, and functional applications with their corresponding references.

MOF	Surface Area (m^2^/g)	Characteristics	Advantages	Disadvantages	Applications	Ref.
**ZIF-8**	~1600	Zn^2+^ + 2-methylimidazolate; sodalite-type	High thermal/chemical stability; simple synthesis	Hydrophobicity limits polar gas capture	CO_2_/CH_4_ separation, membranes, catalysis	[53,54,55,56]
**ZIF-67**	~1500	Co^2+^ analog of ZIF-8; same topology	Redox-active; magnetic; electrocatalytic	Prone to oxidation	Electrocatalysis, batteries, sensors	[57,58]
**UiO-66**	~1100	Zr_6_O_4_(OH)_4_ + BDC; fcu topology	High defect tolerance; stable in aqueous/acidic media	Smaller pore size	Drug delivery, catalysis, CO_2_ capture	[59,60]
**UiO-67**	~2500	Zr_6_ + BPDC linker; larger pores	Accommodates larger guests	Costlier than UiO-66	Guest encapsulation, pollutant adsorption	[61,62]
**UiO-68**	~3300	Zr_6_ + TPDC; very large pores	Superior pore volume; tunable linker	Increased structural defects	Photocatalysis, dye adsorption, enzyme carriers	[63,64]
**MIL-101(Cr)**	~4100	Cr^3+^ + BDC; mesoporous cages	Very high porosity; chemical stability	Chromium toxicity; expensive	Methane/H_2_ storage, catalysis	[65,66]
**MIL-53(Al)**	~1100	Al^3+^ + BDC; breathing framework	Flexible; scalable	Pore instability on activation	CO_2_ separation, VOC adsorption	[67]
**MIL-100(Fe)**	~2200–2900	Fe^3+^ + BTC; MTN topology; mesoporous	Biocompatible; water-stable	Moderate crystallinity	Drug delivery, wastewater treatment	[68,69]
**MOF-5 (IRMOF-1)**	~3800	Zn_4_O + BDC; cubic pores	Historic benchmark; easy to modify	Very moisture-sensitive	H_2_ storage, small-molecule adsorption	[70,71]
**HKUST-1**	~1800	Cu^2+^ paddlewheels + BTC	Open metal sites; strong π-interactions	Moisture instability	CO_2_/CH_4_ storage, catalysis, sensing	[72,73]
**DUT-49**	~5470	Cu^+^ framework; flexible with NGA behavior	Very high surface area; responsive pores	Mechanically unstable	Gas adsorption, negative gas adsorption (NGA)	[74]

**Table 2 polymers-17-01941-t002:** Summary of gas separation performance in various MOF/PIM-based mixed-matrix membranes (MMMs), highlighting the permeability (in Barrer) and selectivity for different gas pairs, along with the corresponding reference number.

Hybrid Material	Type of Gas Separation	Permeability (Barrer)	Selectivity	Ref. nº
CAU-12-ODB/PIM-1	H_2_/N_2_	7199 (H_2_)	127	[129]
ZIF-8/AO-PIM-1	H_2_/CO_2_	5688 (H_2_)	11.97	[139]
ZIF-8/PIM-1	O_2_/N_2_	1287 (O_2_)	3.7	[128]
ZIF-8/PIM-6FDA-OH MMMs	C_3_H_6_/C_3_H_8_ (propylene/propane)	3.5 (propylene)	30	[126]
CALF-20/PIM-1 MMM	CO_2_/N_2_	8003 (CO_2_)	25	[105]
Mg-MOF-74 (CPO-27)/PIM-1 MMM	CO_2_/CH_4_	21,000 (CO_2_)	19.1	[124,146]
Mg-MOF-74 (CPO-27)/PIM-1 MMM	CO_2_/N_2_	21,000 (CO_2_)	15.4	[124,146]
MIL-101-HNO_3_/PIM-1 MMM	CO_2_/N_2_	14,879 (CO_2_)	24.3	[147]
MOF-801/PIM-1 MMM (5% load)	CO_2_/N_2_	9686 (CO_2_)	27	[148]
MOF-808/PIM-1 MMM (2% load)	CO_2_/N_2_	6854 (CO_2_)	23.2	[42]
MUF-15/PIM-1 MMM (optimal 5% load)	CO_2_/N_2_	(38% increase over pure PIM-1) *	17.72	[149]
NH_2_-ZIF-7/PIM-1 MMM (20% load)	CO_2_/CH_4_	2953 (CO_2_)	20.6	[150,152]
agZIF-62/PIM-1 MMM (30% load)	CO_2_/CH_4_	5914 (CO_2_)	67	[151]
ZIF-S/PIM-1 MMM (15% load)	CO_2_/CH_4_	2805 (CO_2_)	21.09	[82]

* In this case, permeability is expressed as the percentage increase relative to the pure PIM-1 membrane.

**Table 3 polymers-17-01941-t003:** Comparison of MOFs, COFs, and MOF–COF hybrids.

Property/Aspect	MOFs	COFs	MOF–COF Hybrids
Composition	Metal ions or clusters linked by organic ligands	Covalent bonds between light elements (C, H, N, O)	Combination of metal ions (MOFs) and covalent organic frameworks (COFs)
Porosity	High, tunable	Permanent, highly ordered	Synergistic, with hierarchical porosity
Thermal/chemical stability	Moderate to high, but limited in harsh environments	High, with chemical robustness	Enhanced stability due to combined frameworks
Electronic properties	Strongly tunable via metal centers	Limited electronic properties	Tunable properties combining MOF’s metal nodes and COF’s organic framework
Functionalization	Broad, via choice of metal ions and linkers	High, via diverse organic chemistry	Enhanced functionalization through integration of metal centers and organic scaffolds
Synthesis challenges	Complex, often requiring specific solvothermal conditions	Moderate, with solvent-based or solid-state approaches	Complex, requiring careful hybridization strategies
Applications	Gas storage, catalysis, sensing	Energy storage, membranes, photocatalysis	Advanced applications: catalysis, gas separation, energy storage, and biomedicine
Advantages of hybrids	-	-	Synergy of MOF and COF properties: high surface area, tunable functionality, stability, and hierarchical pores

**Table 4 polymers-17-01941-t004:** Examples for each type of synthesis of MOF/COF hybrids.

Strategy	Example Material	Application	Reference
MOF-first (in situ COF growth)	NH_2_-UiO-66@TFPT-DETH	Photocatalyst for hydrogen evolution	[183]
MOF-first (amorphous → crystalline COF)	NH_2_-MIL-125(Ti)@COF-LZU1	Photocatalyst for hydrogenation of styrene	[184]
MOF-first (one-step COF synthesis)	NH_2_-MIL-101(Fe)@NTU-COF	Styrene oxidation catalyst	[185]
COF-first (layer-by-layer growth)	COF-300@ZIF-8	Gas separation	[180]
COF-first (sequential MOF-on-COF growth)	Tp-Pa-1 COF@ZIF-8	Photocatalyst for CO_2_ reduction	[186]
Post-synthetic linking	Fe_3_O_4_@A-TpBD@NH_2_-MIL-125(Ti)	EDC absorption and selectivity	[187]

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
