# Peer review of "Metal–Organic Frameworks as Fillers in Porous Organic Polymer-Based Hybrid Materials: Innovations in Composition, Processing, and Applications"

_polymers, 2025, doi:10.3390/polym17141941_

Round 1
Reviewer 1 Report
Comments and Suggestions for Authors
- Section 2 should include a table to clearly compare different MOF structures in terms of their characteristics, physicochemical properties, advantages/disadvantages, and functional applications. The table should include relevant references. Additionally, Tables 1, 2, and 3 are not formatted correctly—please revise them according to the template.
- In Section 2, specific examples of different MOF materials should be provided to illustrate the current research landscape.
- References need formatting corrections: Some journal names are not abbreviated (e.g., Refs. 3, 4, 6, 8, 10, 11…). Please follow the template’s format and remove any URLs.
- For cited works (e.g., Wenbo Chen et al. in line 399, Hang Yin et al. in line 419, Yanan Wang et al. in line 434, Yang Feng et al. in line 455, Chumei Ye and his team in line 474), mechanism diagrams or illustrative figures should be added to clarify the content.
- The figures are not clear enough—Figures 1 and 2 are unreadable. Please replace them with high-resolution images (minimum width/height of 1000 pixels or a resolution of 300 dpi or higher).
- Numbering issues:
- Lines 295–310 should be labeled with numbers (1, 2…).
- The subheading in line 331 should be numbered.
- Lines 778–791 should be labeled with numbers (1, 2, 3…).
- Indentation errors: Lines 736, 743, and 750 lack first-line indentation.
- The discussion on hybrid MOF structures (lines 882–892) should be expanded into a new section, including a summary of the review’s findings and the authors’ insights.
- Tables and figures are not referenced in the text—they should be placed immediately after their first mention in the manuscript.
Author Response
All the answers are coloured in blue
Reviewer 1: Answers
- 1. Section 2 should include a table to clearly compare different MOF structures in terms of their characteristics, physicochemical properties, advantages/disadvantages, and functional applications. The table should include relevant references. Additionally, Tables 1, 2, and 3 are not formatted correctly—please revise them according to the template.
A new table has been included in Section 2 following Reviewer 1 and 3 comments. Tables 1, 2, 3 and 4 are now formatted as indicated in the journal template.
- 2. In Section 2, specific examples of different MOF materials should be provided to
illustrate the current research landscape.
Specific examples have been provided into the text and Table 1 as well.
- 3. References need formatting corrections: Some journal names are not abbreviated (e.g., Refs. 3, 4, 6, 8, 10, 11…). Please follow the template’s format and remove any URLs.
We updated the references format
- 4. For cited works (e.g., Wenbo Chen et al. in line 399, Hang Yin et al. in line 419, Yanan Wang et al. in line 434, Yang Feng et al. in line 455, Chumei Ye and his team in line 474), mechanism diagrams or illustrative figures should be added to clarify the content.
We acknowledge the reviewer’s suggestion to include mechanism diagrams or illustrative figures for the cited works. However, we believe that additional schematic representations are not necessary in this context, as the original references already provide detailed visual explanations. Moreover, these studies are cited to provide background or comparative context and do not constitute the central focus of our work. Including such figures may divert attention from the main theme and objectives of the present study.
- 5. The figures are not clear enough—Figures 1 and 2 are unreadable. Please replace
them with high-resolution images (minimum width/height of 1000 pixels or a resolution of 300 dpi or higher).
Both Figure 1 and Figure 2 are now on high resolution format (enhanced metafile)
- 6. Numbering issues: now lines have changed due to other corrections
• Lines 295–310 should be labeled with numbers (1, 2…).
Corrected
• The subheading in line 331 should be numbered.
Corrected
• Lines 778–791 should be labeled with numbers (1, 2, 3…).
Corrected
- 7. Indentation errors: Lines 736, 743, and 750 lack first-line indentation.
Corrected
- 8. The discussion on hybrid MOF structures (lines 882–892) should be expanded into a new section, including a summary of the review’s findings and the authors’ insights.
A new section has been created, and the requested information has been expanded.
- 9. Tables and figures are not referenced in the text—they should be placed immediately after their first mention in the manuscript.
Now all the Tables and Figures are labeled and referenced in the text. Some of the figures or tables have not been placed immediately after they are mentioned, but rather a little later to ensure that the explanation is clear (just a few paragraphs later).

Reviewer 2 Report
Comments and Suggestions for Authors
This review focuses on the innovative synthesis, performance optimization, and application progress of hybrid materials composed of metal-organic frameworks (MOFs) and porous organic polymers (POPs), systematically expounding on the hybrid systems of various POPs (such as polymers of intrinsic microporosity [PIMs], covalent organic frameworks [COFs], hyper-crosslinked polymers [HCPs], covalent triazine frameworks [CTFs], and conjugated microporous polymers [CMPs]) with MOFs. Although the review of MOF-POP hybrid materials is relatively comprehensive, there is still room for improvement.
1、Some of the figures and tables in the article lack citation annotations. It is advisable to add them at appropriate positions in the main text.
2、The definitions of certain terms, such as Barrer and the Robeson upper bound, can be appropriately supplemented.
3、The analysis of the advantages and disadvantages of methods like the "in-situ growth method" and the "solution mixing method" is not in-depth, and potential risks such as the loss of MOF crystallinity and the degradation of POPs are not mentioned.
Author Response
Reviewer 2: answers
This review focuses on the innovative synthesis, performance optimization, and application progress of hybrid materials composed of metal-organic frameworks (MOFs) and porous organic polymers (POPs), systematically expounding on the hybrid systems of various POPs (such as polymers of intrinsic microporosity [PIMs], covalent organic frameworks [COFs], hypercrosslinked polymers [HCPs], covalent triazine frameworks [CTFs], and conjugated microporous polymers [CMPs]) with MOFs. Although the review of MOF-POP hybrid materials is relatively comprehensive, there is still room for improvement.
- 1. Some of the figures and tables in the article lack citation annotations. It is advisable to add them at appropriate positions in the main text.
Corrected. Some of the figures or tables have not been placed immediately after they are mentioned, but rather a little later to ensure that the explanation is clear (just a few paragraphs later).
- 2. The definitions of certain terms, such as Barrer and the Robeson upper bound, can be appropriately supplemented.
These terms are now explained in a footnote located at the point at which they are first mentioned. The Barrer is a not SI-unit very used to cuantify the gas permeation of single and mixed gases. The Robeson upper bound is the reference-graph used to compare different materials performance, and it has been actuallized two times including new types of materials with enhanced gas permeation properties. More details can be found in the main text.
- 3. The analysis of the advantages and disadvantages of methods like the "in-situ growth method" and the "solution mixing method" is not in-depth, and potential risks such as the loss of MOF crystallinity and the degradation of POPs are not mentioned
These terms are now more detailed and potential risks and degradation issues are now discussed and well referenced.

Reviewer 3 Report
Comments and Suggestions for Authors
The authors should address the following comments:
- The abstract must show the full form for HCPs, CTFs, or CMPs.
- The authors should tabulate the advantages and disadvantages of various MOFs, including ZIF, UiO, MIL, and HKUST.
- The commercial MOFs and COFs, or their hybrids, should be discussed in the setting of domestic applications.
- The authors should include a schematic depiction or visual illustration of the section (“Applications of MOF-PIM MMMs”) to aid readers in comprehending it.
- The synthesis process, surface area, and crystalline structures of hybrid materials should all be included in Table 1. Consequently, discuss about how hybrid materials with surface and structural characteristics are essential for gas separation in the permeability and selectivity processes.
- The schematic diagram's Fig. 4 is not satisfactory.
- The author should provide a table or schematic diagram for the section "4.2. Applications of MOF/COF Hybrids" that shows the approach or concept employed in MOF/COF hybrids for sensors, catalysis, gas adsorption/separation, pollutant removal, and biological applications.
- There was no discussion about Future prospects, challenges, and the review's conclusion.
Author Response
Reviewer 3: Answers
The authors should address the following comments:
• The abstract must show the full form for HCPs, CTFs, or CMPs.
Corrected
• The authors should tabulate the advantages and disadvantages of various MOFs, including ZIF, UiO, MIL, and HKUST.
A new table has been included in Section 2 following Reviewer 1 and 3 comments
• The commercial MOFs and COFs, or their hybrids, should be discussed in the setting of domestic applications.
At present, the applications of MOFs, COFs, and their hybrid materials remain largely confined to research and industrial settings. While significant progress has been made in areas such as pollutant adsorption, catalysis, energy storage and conversion, sensing, biomedicine, and gas separation, these technologies have not yet matured to the point of widespread domestic implementation. Further development in terms of scalability, cost-effectiveness, and long-term stability is required before such materials can be realistically integrated into everyday domestic applications.
• The authors should include a schematic depiction or visual illustration of the section (“Applications of MOF-PIM MMMs”) to aid readers in comprehending it.
A figure has now been included to provide a visual representation of the applications of MMMs. All the applications are listed and represented by icons to help visualize the general versatility of MOF/PIM hybrids.
• The synthesis process, surface area, and crystalline structures of hybrid materials should all be included in Table 1. Consequently, discuss about how hybrid materials with surface and structural characteristics are essential for gas separation in the permeability and selectivity processes.
We appreciate the reviewer’s suggestion regarding the inclusion of synthesis process details, surface area, and crystalline structures in Table 1 (Now Table 2). However, we respectfully note that the hybrid materials investigated in this study are amorphous in nature. As such, they do not exhibit long-range crystalline order, and their surface area characteristics are not the primary factors influencing their gas separation performance. Instead, the permeability and selectivity observed are predominantly governed by the intrinsic amorphous network and the specific chemical functionalities introduced during synthesis. Therefore, we believe that including crystallinity and surface area data in Table 1 would not provide meaningful insight into the structure–property relationships relevant to our findings.
• The schematic diagram's Fig. 4 is not satisfactory.
The schematic diagram’s Figure 4 citation and size (now Figure 5) has been updated.
• The author should provide a table or schematic diagram for the section "4.2. Applications of MOF/COF Hybrids" that shows the approach or concept employed in MOF/COF hybrids for sensors, catalysis, gas adsorption/separation, pollutant removal, and biological applications.
The schematic diagram for section 4.2. “Applications of MOF/COF Hybrids” corresponds now to Figure 5.
• There was no discussion about Future prospects, challenges, and the review's conclusion.
The section has been expanded by completing the information requested by the reviewer.

Round 2
Reviewer 3 Report
Comments and Suggestions for Authors
The auhtors responses to the reviewer's comments are satisfactory.